# Honey’s Antioxidant and Antimicrobial Properties: A Bibliometric Study

**DOI:** 10.3390/antiox12020414

**Published:** 2023-02-08

**Authors:** Christos Stefanis, Elisavet Stavropoulou, Elpida Giorgi, Chrysoula (Chrysa) Voidarou, Theodoros C. Constantinidis, Georgia Vrioni, Athanasios Tsakris

**Affiliations:** 1Laboratory of Hygiene and Environmental Protection, Department of Medicine, Democritus University of Thrace, Dragana, 68100 Alexandroupolis, Greece; 2Department of Microbiology, Medical School, National Kapodistrian University of Athens, 11527 Athens, Greece; 3Centre Hospitalier Universitaire Vaudois (CHUV), 1101 Lausanne, Switzerland; 4Department of Agriculture, School of Agriculture, University of Ioannina, 47100 Arta, Greece

**Keywords:** antimicrobial, antioxidant, honey, VOSviewer, bibliometric, R bibliometrix

## Abstract

Research attention has been drawn to honey’s nutritional status and beneficial properties for human health. This study aimed to provide a bibliometric analysis of honey’s antioxidant and antimicrobial properties. The research advancements within this field from 2001 to 2022 were addressed using the Scopus database, R, and VOSviewer. Of the 383 results, articles (273) and reviews (81) were the most common document types, while the annual growth rate of published manuscripts reached 17.5%. The most relevant topics about honey’s antimicrobial and antioxidant properties were related to the agricultural and biological sciences, biochemistry, and pharmacology. According to a keyword analysis, the most frequent terms in titles, abstracts, and keywords were honey, antimicrobial, antioxidant, bee, propolis, phenolic compounds, wound, antibacterial, anti-inflammatory, and polyphenols. A trend topic analysis showed that the research agenda mainly encompassed antioxidants, pathogens, and anti-infection and chemical agents. In a co-occurrence analysis, antioxidants, anti-infection agents, and chemistry were connected to honey research. The initial research focus of this domain was primarily on honey’s anti-inflammatory and antineoplastic activity, wound healing, and antibacterial agents. The research agenda was enriched in the subsequent years by pathogens, propolis, oxidative stress, and flavonoids. It was possible to pinpoint past trends and ongoing developments and provide a valuable insight into the field of honey research.

## 1. Introduction

Honey is a natural product produced by bees. It has been used since ancient times as a food and in various therapeutic applications. It contains several chemical components beneficial to human health, which depend to a large extent on the botanical origin of the plants and the geographical area. Moreover, those two characteristics are used for certification and authenticity. Thus, a significant variability is observed in the physicochemical properties of honey, such as the moisture, pH, free acidity, and electrical conductivity, as well as the content of chemical substances such as organic acids, proteins, amino acids, vitamins, and phenolic compounds. For many years, honey and other beekeeping products such as propolis, pollen, royal jelly, bee venom, and wax have been recognized as bioactive products, with their therapeutic effects extending to a wide range of diseases and infections by pathogens, parasites, and viruses. The action of honey and beekeeping products is mainly due to antioxidant, anti-inflammatory, and immunomodulation factors [1,2,3,4,5,6,7]. Figure 1 illustrates several of the primary bee products and the most prevalent flavonoids and phenolic compounds available in honey and other bee products. The principal therapeutic properties from the respective bioactive compounds are also documented. Variability of the chemical composition and synthesis should be considered due to honey and honey products’ varied botanical and geographical origins [8,9,10,11,12,13,14,15,16,17,18].

Honey research has also extended to more contemporary and complex medical issues. In nutrition science, the consumption of honey and its association with the gut microbiome has recently been studied, since the gut microbiome plays an essential role in chronic diseases. The presence of honey polyphenols may enhance alterations in the balance between pathogenic and beneficial microbial populations in the gut microbiome, providing a beneficial effect [19].

Polyphenols of honey and beekeeping products were also studied for their role in strengthening the human immune system, concerning their contribution to the mitigation of COVID symptoms and the reduction of patient recovery periods. Supplementation of honey with standard pharmaceuticals, such as opinavir/ritonavir tablets, arbidol, chloroquine phosphate, hydroxychloroquineoroseltamivir with or without azithromycin, chloroquine or hydroxychloroquine, or oseltamivir corticosteroids showed promising outcomes in various clinical studies [20].

In addition to polyphenols, honey also contains several carbohydrates, specifically oligosaccharides, and a beneficial role was highlighted in terms of their consumption. Non-digestible oligosaccharides have been studied for their prebiotic effect on the gut microbiome. Specific types of honey may benefit gut microbial populations, offering various nutritional benefits to human health, related to reducing infection and inflammation, as well as obesity, only in animal models [21,22].

Currently, bibliometric analyses for honey in the Scopus database are rare. By entering the keyword “honey” and scanning the database for bibliometric studies, 13 review documents and bibliometric analyses were collected. Six review studies referred to honey, specifically to applications of beekeeping products, their clinical trials and the results of their applications, to bee therapy in general, and the antioxidant properties of beekeeping products in relation to the environment and medicinal plants. The cultural significance of bees in human culture, the presence of chemical contaminants in honey products, and finally, the treatment of diabetic foot ulcers by dressing with honey were also investigated [23,24,25,26,27,28].

The existing bibliometric studies, of which only three were conducted using the CSIRO database, investigated various topics around honey and bee products, such as the use of beekeeping products in medicine or the honey itself from 2011 to 2020, honey consumption, and the collapse phenomenon of bee colonies. Likewise, pesticide presence in bee products and honey, the mortality of bees, as well as issues of authenticity and the development of technological methods to determine the origin of honey were studied [9,29,30,31,32,33]. From the above, it can be concluded that the antimicrobial action of honey combined with its antioxidant properties has yet to be extensively and scientifically studied over a long period.

The aim of this research was the scientific records of honey’s antimicrobial and antioxidant properties in the Scopus database, from its inception and over the entire available period, as well as mapping of the research frontiers and the research trends in the above topics with modern bibliometric techniques. Additionally, with the help of bibliometric indicators, it is possible to capture the evolution of a scientific object over the years, its research extensions, and the new methods and research protocols it has employed, as well as the revision and integration of further information and fields. Finally, this is the first time that the topic of the antimicrobial activity of honey has become a research theme for bibliometric analysis.

The following research questions (R.Q.) offer academics and decision-makers a perspective on the content and topics addressed in the literature. RQ1: What revisions have occurred in the literature on the antimicrobial activities of honey? RQ2: What have been the most influential research articles, authors, and manuscripts from those published in Scopus? RQ3: What are the most important topics discussed in the research field of honey, concerning its antioxidant and antimicrobial properties?

This study contributes to the scientific literature, while the bibliometric indexes of this research outline the research production on the antimicrobial and antioxidant properties of honey. Scientific manuscripts cover the period up to the present literature (till 12 November 2022), when research attention on honey and its health benefits was expanded into the nutrition and pharmaceutical domains. Thus, this research aimed to assess the trends and frontiers in the specific health benefits of honey using the Scopus database, various bibliometric indicators, and visualizing the respective scientific literature. An additional goal of this bibliometric study was to reveal the disciplines engaged with honey’s antimicrobial and antioxidant activity, which topics appeared over time, and how they evolved. Moreover, highlighting the primary research streams and presenting the current achievements, ongoing challenges, and trending issues is also accomplished with the aid of bibliometric tools.

## 2. Materials and Methods

For the above research, a search was performed in the Scopus database. It includes 1.7 billion cited references and covers nearly 2500 serial titles from approximately 7000 publishers in top-level subject fields: life, social sciences, physical sciences, and health sciences. Additionally, the Scopus database offers many advantages such as diversification, the flexibility of research fields, and an advanced document analyzer mechanism. This mechanism is based on Boolean Syntax for retrieving documents using combining keywords with various Boolean operators [34,35,36,37]. The Scopus database provides many scientific journals and an advanced indexing operation [38].

After a preliminary application of various combinations, the phrase “antimicrobial antioxidant honey” was used, with a time range from the initial date of Scopus to 12/11/2022 and with language selection in English, and the following search details: TITLE-ABS-KEY (antimicrobial AND antioxidant AND honey) AND (LIMIT-TO (DOCTYPE, “ar”) OR LIMIT-TO (DOCTYPE, “re”) OR LIMIT-TO (DOCTYPE, “ch”) OR LIMIT-TO (DOCTYP, “cp”) OR LIMIT-TO (DOCTYPE, “cr”)) AND (LIMIT-TO (LANGUAGE, “English”)). Research documents, reviews, manuscripts from conference proceedings and conference reviews, and book chapters were included.

Moreover, the obtained manuscripts were recorded in the Microsoft Excel program by year, subject area, document type, and institutional affiliation. Visual depiction of keywords plus was also realized in this bibliometric research, and specifically, word dynamics and word trends of keywords plus are illustrated. The VOS Viewer program was applied to visualize the results and create a bibliographic map. We performed a co-authorship analysis using the full counting method, assigning the same weight to each co-authorship link. The full counting method was further used in the co-occurrence analysis of the keywords in the manuscript title, abstract, and text. The bibliometric analysis was developed by executing the following steps: research criteria, study questions, and analysis approach selection (the year, subject area, document type, institutional affiliation, keyword analysis, network of authors, research evolution); bibliometric data selection and analysis using the bibliometric software; and through generating networks, visualization figures, and interpretation of the results (Figure 2).

## 3. Results and Discussion

Research articles (273) and reviews (81) were the most common document types published for “antimicrobial and antioxidant and honey” of the 383 manuscripts (Table 1).

The top ten institutions that published research documents for the abovementioned keywords were also identified. The Università Politecnicadelle Marche and the King Abdulaziz University were the most productive institutions, followed by the University of Rzeszów, the Karadeniz Technical University, and the University of Belgrade-Serbia. Moreover, five European countries, Saudi Arabia, Turkey, Brazil and Thailand, were ranked among the top ten productive research organizations (Table 1).

Considering the top ten productive authors, only Giamieri F had a two-digit number of published documents, namely 11. The number of manuscripts produced by the other authors ranged from five to nine. Battino M. had the best score, with a 70 h-index from the Università Politecnicadelle Marche, Ancona, Italy. In contrast, Oses, S.M., with a 16 h-index from Universidad de Burgos, Burgos, Spain, was the author with the lowest score. The h-index of the authors is also documented in Table 1.

Figure 3 illustrates the annual production of documents. The number of published manuscripts represents a plausible assessment of the trends in a specific research domain. Publications were categorized by year, uncovering that the first manuscript appeared in 2001; however, an exponential growth in the produced articles was seen after 2009 (Figure 3). The highest number of manuscripts (68) was published in 2022. The annual growth rate was 17.5%. Furthermore, the yearly publication graph shows that researchers started paying attention to honey’s antimicrobial and antioxidant properties and various issues in 2011. The scientific interest is evident, because there has been an exponential increase in published documents since 2011.

Two manuscripts appeared in 2001 and are considered the initial efforts to document honey’s antimicrobial and antioxidant properties indexed in the Scopus database. The first research paper [39] highlighted the inhibitory activity of honey against foodborne pathogens, concerning floral sources and emerging antioxidant properties. The second document [40] discussed honey’s vital role in wound healing and the absence of side effects in the healing process. Although published over 20 years ago, these works still address ongoing research patterns regarding honey’s beneficial properties.

The list of the ten most productive countries includes countries from five continents. Asia had in total of 120 documents, namely: India (32), Turkey (31), Malaysia (30), and Saudi Arabia (27). Italy (28), Poland (28), and Spain (25) represented Europe with 81 records, while Egypt (21) was the only country from Africa. The top ten most productive countries list includes USA and Brazil, with 22 and 26 published manuscripts, respectively. Figure 4 highlights the number of documents that combine the subject area of honey products and the antimicrobial and antioxidant properties research theme.

The list of the most productive countries in terms of their research performance partially coincided with the most productive countries in terms of honey production. Countries such as Turkey, Brazil, India, and America are in the top ten honey producers, and Spain is in 13th place (https://www.nationmaster.com/nmx/ranking/honey-production-fao (accessed on 15 January 2023)). In the present bibliometric analysis, some countries lagged behind in honey production even though they were on the list of high research production in absolute numbers of documents, such as Italy, Poland, S. Arabia, and Egypt.

Thus, diversification was observed in the research output and honey production between countries. Honey research can focus on various technological innovations and research directions: antioxidant and antimicrobial properties, medical applications, honey products, entomology and bee biology, product standardization and marketing, and botanical and geographical origin can explain the detected research pattern.

For example, the research production of S. Arabia focused on the whole spectrum of honey’s antimicrobial and antioxidant activity, as well as on the therapeutic properties of honey and bee derivative products, such as propolis. This research spectrum was observed in the three articles with the most citations [41,42,43]. Italy’s research on honey consisted mainly of its beneficial properties concerning its chemical and phenolic composition and the therapeutic properties of honey products [14,15,44]. Moreover, the biological properties of melanoidin produced in foodstuffs, such as honey, and the antibacterial and antioxidant agents of various types of Polish honeys were the subject of the three most cited papers that emerged from Poland [45,46,47]. The research activity of Egypt covered the flavonoid compounds of honey, such as pinocembrin and their respective pharmacological and biological properties. In addition, using honey and bee products against COVID-19, and the phenolic, antioxidant, and antimicrobial characteristics of Egyptian honeys were topics included in the research production of the only African country on the list of the most productive countries [48,49,50]. 

Regarding the bibliometrics, the collaboration metrics and indicators mainly included co-authorship pairs of countries, universities and institutions, and authors. These metrics aim to describe patterns representing the total scientific impact and influence of international collaboration. However, elucidating collaboration metrics is challenging, and additional metrics should also be considered, to produce a visualization map of collaboration networks [51,52].

Global collaborations, networks, and research streams among countries are illustrated in the following figure. Figure 5 represents a clustered network of countries’ collaborations in a circular shape. The thickness of the lines is proportional to the number of collaborations between countries. Moreover, collaboration between two countries resulted from the co-authorship of authors from two different countries. Thus, the network node size is proportional to the number of publications in that country. In addition, the number of countries illustrated is twenty. In the current bibliometric analyses, single-authored documents totaled 22, co-authors per manuscript came to five, while international co-authorships amounted to 26.63%.

Five clusters can be recognized, including countries from at least two continents. The first cluster is dominated by Portugal and Brazil (red cluster). Greece, Serbia, and Poland also belong to this cluster. The second cluster (blue color) includes three countries, Turkey, Iran, and Italy. The green cluster encompasses more countries, namely, eight: Australia, Malaysia, Pakistan, Egypt, USA, India, Saudi Arabia, and China. The purple cluster has three countries, Spain, Italy, and Ecuador. Only one country, Argentina, exists in the fifth cluster (orange).

Notably, the above five countries dedicated significant research efforts to chemistry and life sciences from October 2021 to September 2022, as indicated in the index produced by Nature about the collaboration of countries. In accordance with the Scimago Journal & Country Rank regarding the country comparison, it was appropriate to consider additional metrics for collaboration and international research initiatives stemming from these countries. Thus, Italy had the biggest h-index 1.2 k, followed by Brazil (h index 690), Portugal (h-index 559), Saudi Arabia (h-index 478), and Egypt 349(h-index). Regarding the percentage international collaboration, in the last two years, Saudi Arabia was the leading country, followed by Egypt, Portugal, Italy, and Brazil. Since 2009 all these countries have increased their collaboration networks and international research streams (https://www.scimagojr.com/comparecountries.php?ids[]=sa&ids[]=eg&ids[]=pt&ids[]=br&ids[]=it (accessed on 15 January 2023)).

Figure 6 highlights the number of documents that combined the subject area of honey products and the antimicrobial and antioxidant properties research theme.

These properties of honey have become a topic for multiple disciplines beyond the agricultural and biological sciences. More than 20 subject areas cover this scientific domain. Most of the retrieved manuscripts belonged to agricultural and biological sciences (23.2%); biochemistry, genetics, and molecular biology (13.7%); pharmacology, toxicology, and pharmaceutics (12.9%); and medicine (12.4%). Furthermore, as shown in Figure 6, chemistry, immunology, and microbiology were also linked to antioxidants, antimicrobials, and honey.

Considering the relevant sources, the scientific journals that published a double-digit number of articles included *Molecules* (15), *Antibiotics* (12), *Lwt* (12), *Food Chemistry* (10), and *Foods* (10). The list of the most relevant journals included *the Journal of Agricultural Research* (10), *Applied Sciences* (6), *Evidence Based Complementary and Alternative Medicine* (6), *Food and Chemical Toxicology* (6), and *Food Research International* (6).

The above scientific journals have a high scientific impact and cover various multidisciplinary research topics. In particular, the lowest citation score was 3.2 (2021) for *Evidence-based Complementary and Alternative Medicine*, and the highest was 13.1 (2021) for the journal *Food Chemistry*. The many research areas of these sources include organic chemistry, medicinal chemistry, natural products, natural antibiotics, bee management, and advances in research into antibiotics and related bioactive therapeutic agents.

The top ten most cited articles referring to the antimicrobial and antioxidant properties of honey are listed in Table 2. The highest citation number was 660 for the manuscript entitled “Honey for nutrition and health: A review”. This document received the most outstanding scientific attention and was published in 2008 in *the Journal of the American College of Nutrition* [53]. It is worth highlighting that the top ten most highly cited papers were published in various journals from 2001 to 2018.

The most cited document addressed honey’s composition and nutritional contribution, while the second highest discussed the antioxidant and antimicrobial activities when utilizing reactive oxygen species [54]. The third article dealt with the health effects, biochemical composition, and functional properties of honey and products such as propolis and royal jelly [17]. The botanical origin, chemical composition, and biological properties of various types of honey are considered in the last two documents of the ranking. Notably, the latest published date of the five most cited documents was 2013 [55,56].

Continuing with the other most influential documents, we identified manuscripts that focused on correlating the botanical origin of honey with characteristics such as its color, total phenolic, flavonoid, ascorbic acid, amino acid, protein, and carotenoid contents. Honey’s antimicrobial capacity was also tested against Gram-positive and -negative bacteria, showing miscellaneous sensitivity results. The correlation between the chemical components of honey, such as phenolic content and hydrogen peroxide, influenced the general antimicrobial activity of the honeys, as documented in the following article from the list presented in Table 2. Dark-colored honeys achieved an increased inhibitory effect compared to lighter-colored honeys of various pathogenic microorganisms, such as Escherichia coli O157:H7, Salmonella typhimurium, Shigella sonnei, Listeria monocytogenes, and Staphylococcus aureus [39,44].

The following article presented an extensive literature review of the beneficial properties of honey in human health and recorded, in addition to its known antimicrobial and antioxidant properties, beneficial effects on cardiovascular function, the respiratory system, the fight against cancerous tumors, and the fight against diabetes [15]. Subsequently, the therapeutic properties of honey and beekeeping products, the characteristic chemical and bioactive substances, and their uses in traditional medicine were the research objectives of the ten most influential articles [14,57].

In the present bibliometric analysis, four articles also used a questionnaire. The first survey investigated the acceptance of an innovative product consisting of honey and propolis among 69 consumers. The content of the above product with honey and propolis at 0.5% was characterized as unpleasantly resinous and bitter. Finally, the composition at rates of 0.5% and 0.3% of propolis were chosen as suitable potential combinations for commercial exploitation [58]. A similar study on the organoleptic characteristics of honey with added spices found that they impacted honey’s taste and aroma, with the combination of honey and cinnamon being the most acceptable to consumers [59]. One survey included questions about awareness of alternative cough remedies, including the use of honey. Of the participants, 39 out of 40 signified their satisfaction with applying honey to treat their child’s cough. In addition, the factor of parents and child age and gender did not seem to affect the cough assessment score [60]. Finally, a questionnaire was distributed to 32 beekeepers with gender, age, level of education, and basic information about their beekeeping activity, such as the number of hives, the amount of honey produced, etc., in North-Central Morocco. In addition, the participating beekeepers were asked to provide information on the botanical diversity of the area regarding melliferous plants. The objective of the study was taking an agronomic approach to the area, in relation to beekeeping and its further economic and agronomic development [61].

Figure 7 illustrates the network of authors and the number of documents based on bibliographic coupling. The principle of the bibliographic coupling method is the exploitation of the cited documents shared between papers for building relationships and generating a network of authors, documents, institutions, and countries. Thus, bibliographic coupling considers citations, to reveal similarities between items, such as two documents, authors, institutions, or countries [62]. Bibliographic coupling can be considered a measure of intellectual relatedness and research frontier composition [63,64].

Subsequently, when two manuscripts cite a third, they have a relationship and should be grouped in a visualization map. The bigger the node size (author), the more productive the author. The distance between the nodes in the visualizing network is proportional to their subject-relatedness. Node distance in the network map represents their subject relatedness. The thickness of the lines between the network nodes illustrates the bibliographic coupling strength between the authors [64].

Author’s scientific links and international collaboration have mutual benefits for the productivity of the authors, for the global impact of their institutions, and for the scientific standing of the countries. Thus, international authorship and diversification of affiliations in research papers can elevate authors’ scientific impact and increase their publishing activity and prestige [65].

The minimum number of manuscripts per author was set as four. Therefore, only 16 of 1730 authors were considered for inclusion in the visualization network. In particular, three clusters are shown, the blue with three authors (cluster 3), the red with seven authors (cluster 1), and the green with six authors (cluster 2). The respective number of documents for each group was 13, 34, and 43. Osés, Sandra M was the leading author for total link strength (2606) in the red cluster. His research focused on honey, botany, stingless bees, spray drying, and drug formulation [58,66]. Giampieri Francesca was ranked topmost in the red cluster. She received a 5067 total link strength and was scientifically active in antioxidants, wound healing, honey, and botany [11]. Fett Roseane was the most influential author in the blue cluster, with 1869 total link strength. Her research topics were honey, bees, antioxidants, filters, chitosan, and botany [67].

The following figure (Figure 8) corresponds to the word clouds of the 50 most frequent keywords in title and abstract manuscript fields.

The above word clouds show that the most frequent keywords occurred more than ten times in titles, abstracts, and keywords. There was a high level of similarity among the most frequent words: honey, antimicrobial, antioxidant, bee, propolis, and phenolic compounds. “Pollen”, “wound”, “antibacterial”, “anti-inflammatory”, “chemical”, “polyphenols”, and “manuka honey” were also words that appeared in the collected manuscripts. As expected, the search keywords antimicrobial, antioxidant, honey; the respective stemming terms; and their derivatives prevailed in the word clouds.

Research into the properties of manuka honey began in the early 90s and has continued ever since. Studies on the beneficial properties of this honey have expanded in many areas of medicine, pharmacology, and immunology. In addition, the antimicrobial activity of manuka honey and its mechanism of action have been investigated against various pathogens, such as *E. coli*, *Clostridioides difficile* strains, and *Mycobacterium abscessus* [68,69,70,71,72,73,74]. 

In the following figures, word dynamics and trends of Keywords plus are depicted. Keywords plus are produced automatically and include words or phrases derived from an article’s citation list, yet they do not necessarily exist in the title. Furthermore, Keywords plus provide a detailed picture of a manuscript’s scientific content and research depth, and their implementation in bibliometric analysis has become popular [75].

The dynamics of ten keywords since 2001 are represented in Figure 9. The specific terms are ranked in ascending order: “honey, antioxidant activity, nonhuman, article, antioxidants, antioxidant, antimicrobial activity, *Staphylococcus aureus*, *Eschericia coli*, and anti-infective agent”.

The term with the most significant dynamic and an escalating upward trend in terms of its use was “honey”, which was also a search term. In the year 2001, it appeared three times, and in the year 2022, it occurred 291 times. For the rest of the terms, it is noticeable that until the year 2009, they all had single-digit appearance numbers, while recent years ranged from 87 (anti-infective agent) to 133 (non-human). The dynamics of terms related to honey’s microbial and antioxidant properties also showed a significant increase in their usage. In addition, the dynamics of terms referring to microorganisms were also shown to be escalating. It is also noteworthy that among the most dynamic terms were Gram-negative bacillus (*Escherichia coli*) and Gram-positive coccus (*Staphylococcus aureus*), which also shows the research diversity recorded concerning pathogenic microorganisms [43,76,77,78].

Selecting keywords plus one can reveal the main topics and research trends. As a result of the trend topic analysis, it was observed that the research on honey’s antimicrobial and antioxidant activity mainly concentrated on antioxidants, pathogens (*E. coli*, *S. aureus* and *P. aeruginosa*), anti-infective, and chemical agents, as well as on propolis [42,79] (Figure 10).

Notably, through a trend topics analysis, the years in which the focus topics obtained through keyword plus analyses became essential were revealed. This technique can provide scientific data on study areas, trends, and the information that developed from these trends annually. According to the analyses, it was noteworthy that in the present decade, 2020–2022, the terms “gentamicin, probiotic agent, and radical scavenging assay” have started to appear in publications. Trends in essential concepts such as research on *H. pylori*, anti-bacterial agents, fructose, and *C. albicans* can be said to be attracting great interest, since these topics have been investigated continuously, with a minimum of eight years of research. Furthermore, the antimicrobial activity of honey, one of the vital concepts concerning pathogenic microorganisms such as *E. coli*, *P. aeruginosa* and *S. aureus*, emerged in 2013 and sustained its prominence until 2021 [80,81,82,83].

The high prevalence of these microorganisms in Keywords plus and terms such as “anti-ineffective agents, propolis, animals, anti-inflammatory activity, antibacterial agents, and hydrogen peroxide” emerging from the current literature emphasize the important role played by these research topics. A particular emphasis was laid on critical studies concerning honey properties such as sugars (glucose and fructose), color, polyphenols concentration and identification, flower properties, and origin, as well as antibacterial agents such as gentamicin and chitosan [41,84,85,86].

Figure 11 illustrates a network visualization map of the co-occurrence of keywords. This is based on the 4129 terms extracted from the retrieved titles, keywords, and abstract fields of the selected documents. Setting the minimum number of keyword occurrences to 20, only 55 met the threshold.

The map categorizes terms into two significant clusters, with respective colors. The color of an item is determined by the cluster to which it belongs. Moreover, the closer two items are located on the map, the stronger their relatedness. Figure 11 represents the two cluster items with the best score for total link strength. Moreover, total link strength measures each keyword’s significance through an assigned weight in VOS software. As such, antioxidant and antimicrobial activity are the items in the first cluster (red color) with the best score, confirming the essential search terms. Honey, anti-infective agents, and antioxidants belong to second cluster (green), with the best score for total link strength (Figure 11).

Based on the network map, two clusters were identified. The terms with the highest occurrence from the first cluster were antioxidant activity, antimicrobial activity, antioxidant, nonhuman, human, and controlled study. Similarly, the terms from the second cluster were honey, antioxidant, anti-infective agent, chemistry, and phenol derivative. An analytical description of the correspondence terms is presented in Table 3.

The red cluster contains 34 keywords and their co-occurrence relationships. This cluster delimits the research frontiers of honey’s antimicrobial, antifungal, and antioxidant properties. It contains terms such as controlled study, minimum inhibitory concentration, hydrogen peroxide, quercetin, ascorbic acid, gallic acid, and enzyme activity, as well as three pathogenic microorganisms of significant importance: *Staphylococcus aureus*, *Escherichia coli*, and *Pseudomonas aeruginosa*. Moreover, this cluster includes additional terms such as chemical composition, radical scavenging assay, natural product, propolis, polyphenol, antineoplastic activity, plant extract, physical chemistry, oxidative stress, and HPLC. The research topics of honey, wound healing, and antineoplastic activity are also found in this cluster.

The green cluster includes 21 keywords and mainly refers to honey’s chemistry and chemical compounds. Chemistry, phenol derivative, food products, animals, phenols, anti-bacterial agents, flavonoids, drug effect, phenolic compounds, polyphenols, pollen, microbial sensitivity test, and anti-inflammatory agent are keywords residing in this cluster. The second cluster covers the main anti-ineffective, antimicrobial, anti-bacterial, and anti-inflammatory issues. This cluster provides research approaches to anti-bacterial and anti-inflammatory agents and considers the relationship between honey and the mode of action of chemical compounds. 

Methodical and extensive research has been carried out into the properties of manuka honey. Its proven utility in antimicrobial and therapeutic applications has made it a valuable solution to various medical problems. The superiority of manuka honey lies in its wide variety of flavonoids and multiple chemical compounds, which increase its therapeutic effect [87]. Honey’s hydrogen peroxide (H_2_O_2_) content significantly influences its antimicrobial activity. In addition to H_2_O_2_, methylglyoxal (MGO) also affects honey’s antimicrobial activity. In manuka honey, MGO ranges from 38 mg/kg to 761 mg/kg, which is, comparatively, up to 100-times higher than most common types of honey [75]. The antimicrobial activity of manuka has been confirmed in microbial cultures grown in vitro aerobically and anaerobically, such as *Staphylococcus aureus*, *Pseudomonas aeruginosa*, *Salmonella typhimurium*, and *Enterococcus faecalis* [77,88,89].

Manuka honey has also been studied regarding wound healing and tissue regeneration. In addition to its antimicrobial action, it prevents the formation of microbial biofilms and reduces the infected area. Its action in tissue healing and renewal is based on reducing inflammatory cell response and, concurrently, activating cytokine production in the wound area, resulting in the proliferation of epithelial cells. On this basis, various formulations based on honey, and in combination with other substances such as hyaluronic acid and zein coatings, have been tested [90,91,92,93].

The antimicrobial activity of honey and its research significance was established by the corresponding keywords found in both clusters, highlighted by the bibliometric analysis. Additionally, in the red cluster, there are indications for honey research regarding its antifungal activity: “antifungal activity, *Candida albicans*” [82,94]. Moreover, the research activity on honey’s antimicrobial and antifungal properties not only focused on the types of honey, based on their geographical and botanical origin, but also products derived from honey [12,95,96,97,98,99].

The presence of the terms “high-performance liquid chromatography, in vitro study’’ in the red cluster refer to various technological research aspects. The link between the beneficial properties of honey and beekeeping products focuses on characterizing these antimicrobial and antioxidant substances. Methodological approaches using specific instruments, such as liquid chromatography, are recorded in the literature. Such studies emphasized the determination of the chemical components of antimicrobial and antifungal activity, the identification and authentication of honey varieties, and the determination of the geographical origin and the corresponding chemical profile [56,98,99,100,101,102,103,104,105,106,107].

The terms “quercetin, ascorbic acid, and gallic acid”, appearing in the same cluster as the term HPLC, are chemical substances used as standards for determining bioactive compounds such as total phenolic content, phenolic compounds, antioxidant activity, vitamin C, and sugars. Together with the other terms that appear in both clusters (“natural product, polyphenol, chemical composition, phenols, anti-bacterial agents, flavonoids, polyphenols’), they depict the research output determining the phenolics and flavonoids in honey, as an essential component of the verification of their antimicrobial and antioxidant activity [108,109,110,111,112,113].

In-depth research has also been conducted on other beekeeping products, such as propolis. This natural product has attracted the attention of scientists for decades because of its chemical composition, which offers many therapeutic utilizations and a multitude of pharmacological applications. Propolis chemical synthesis is commonly composed of 50% resin, 30% wax, 10% essential oils, 5% pollen, and 5% other substances, which include organic compounds and minerals such as phenolic acids (cinnamic and caffeic acid) or their esters, terpenes, flavonoids (flavones, flavanones, flavonols), aromatic aldehydes, alcohols, and fatty acids among others. Propolis’ chemical composition is related to the geographical, botanical, and environmental conditions of the region of harvest [114,115,116].

Propolis has been studied in combination with honey and other substances, such as nanomaterials, to restore, regenerate, and heal skin damage and burn wounds. It has also been analyzed for its use as a bio preservative and food additive, as well as for its antimicrobial and antioxidant properties against pathogenic microbes and fungi such as *Salmonella enterica*, *Yersinia enterocolitica*, *Staphylococcus aureus*, and *Listeria monocytogenes*; *Propionibacterium acnes*; and the fungi *Candida albicans* and *Saccharomyces cerevisiae* [57,117,118,119,120,121].

In the second cluster, the green one, the keywords “bee” and “pollen” also appear. They are related to honey research and beekeeping. The stingless bee and its products, honey, propolis, and pollen, have attracted particular attention in the beekeeping research community. These products are being studied for their beneficial effects, not only from a medicinal point of view, but also from a nutritional point of view, with foods containing stingless bee honey being considered a super-food [122,123].

Pollen is also highlighted in the green cluster. Processed pollen, which acts as a nutritional source for honey bees, is called bee bread. This honey product is a potential functional food with research initiatives identifying over 300 bioactive molecules from various geographical areas worldwide, such as free amino acids, fatty acids, minerals, vitamins, organic acids, and polyphenols [18,124]. Research on the properties of pollen and the combination of pollen with other bee products reflects the research on its antioxidant, antiradical, antibacterial, antifungal, physical, chemical, anti-inflammatory, and pharmacological characteristics [125,126,127,128,129].

A visualization of the selected keywords and the evolution of research using the total link strength of VOS viewer is shown in Figure 12. The research trends of the last five years, 2017–2021, are also given. The initial research focus in this domain was primarily on honey’s anti-inflammatory and antineoplastic activity, wound healing, and antibacterial agents.

Identifying potential therapeutic compounds in honey and the antioxidant background are topics that emerged in the scientific community. The use of honey as a therapeutic agent for various diseases and its role in controlling disease conditions have gained attention. Research papers revealing honey’s mode of action based on flavonoid and polyphenol substances and the antimicrobial spectrum were also produced by various authors. Honey and honey product investigation emphasized the antitumor activity and anti-inflammatory capacity of propolis and the positive biological effects in medicine and the cosmetic industry [14,130,131,132,133].

In the subsequent years, the research agenda was enriched by *E. coli*, *S. aureus*, propolis, phenolic compounds, oxidative stress, and flavonoids. Research issues concerning anti-infection and anti-inflammatory agents were also highlighted. At the same time, pollen and free radical scavenging also entered the scientific domain of the antimicrobial and antioxidant usage of honey.

Scientific works have already demonstrated the antimicrobial status of propolis and the combination of propolis with honey. Moreover, pollen was investigated in various clinical projects, probing its radio-protective, anti-chemo toxicity, and nutrition potential. Research concerning propolis mainly concentrated on botanical origin, flavonoid content, and antibacterial effectiveness, due to cytoplasmic membrane function modification, inhibition of biofilm formation, and cell membrane proteins. Scientific studies have already demonstrated the antimicrobial status of propolis and the combination of propolis with honey. Moreover, pollen was investigated in various clinical projects probing the radio-protective, anti-chemo toxicity and nutrition potential. Research concerning propolis mainly circulated to the botanical origin, flavonoid content and antibacterial effectiveness due to cytoplasmic membrane function modification, inhibition of biofilm formation and cell membrane proteins [42,134,135].

Bacterial antimicrobial resistance to pathogens was another topic that emerged in the literature regarding the clinical aspects of honey and its derivative products. The role of reactive oxygen species (ROS) in antimicrobial resistance was also highlighted. ROS in the form of H_2_O_2_ and oxidative stress significantly damages DNA, RNA, and cell permeability [135].

In addition, antibacterial tests with honey against significant pathogens such as *E. coli* and *S. aureus* were conducted between 2018 and 2020. In this regard, the antibacterial activity of honey against methicillin-resistant (MRSA) *S. aureus* isolated from patients with diabetic foot ulcers was recorded in 2020. The authors concluded that honey could be efficiently used for diabetic foot ulcer treatment, to combat *S. aureus* infection. Honey type is a crucial factor determining the levels of phenolic and flavonoid compounds and, thus, different antibacterial behaviors. In dark forest honey and honeydew honey, these compounds performed successfully against Gram-positive and Gram-negative bacteria [136,137,138,139,140,141].

Furthermore, the biological activity and the role of gallic acid and quercetin entered the field at the end of the last decade, along with the bioactive properties of antioxidants, and new determination and identification methodologies. Determination of phenolic compounds using high-performance liquid chromatography (HPLC) utilizes gallic acid, caffeic acid, syringe acid, chlorogenic acid, p-coumaric acid, ferulic acid, catechin, quercetin, and chrysin, among others.

These compounds characterize the color, sensory profile, and therapeutic properties of honey. Total phenolic acid and flavonoid content, hydrogen peroxide removal activity, the effect of removal of DPPH radicals, total antioxidant capacity, and measurement of reduction potential are representative methods applied in honey research, to emphasize its nutritional and health-beneficial activities. Concerning the physicochemical properties of honey, specific emphasis was given to color, water and moisture content, organic acids, acidity and pH, electrical conductivity, and ash content [99,105,140,141,142].

## 4. Conclusions

Honey and honey products are of great importance in terms of their nutritional and therapeutic properties, as well as the health implications for the hygiene status of humans. Various research projects and initiatives have focused on the antimicrobial and antioxidant profiles of various types of honey. Honey is in the spotlight of worldwide research activity, and this activity has expanded beyond the botanical origin of honey and its products. Honey research encompasses clinical studies and multiple pharmacological perspectives.

This bibliographic analysis revealed the antimicrobial and antioxidant attributes of honey explored since 2001. Three-hundred-eighty-three manuscripts met the search criteria in the Scopus database. The most retrieved manuscripts belonged to the agricultural and biological sciences, biochemistry, genetics and molecular biology, pharmacology, toxicology, pharmaceutics and medicine. The list of the ten most productive countries included countries from five continents. Asia had a total of 120 documents, namely: India (32), Turkey (31), Malaysia (30), and Saudi Arabia (27). Italy (28), Poland (28), and Spain (25) represented Europe with 81 records, while Egypt (21) was the only country from Africa.

Subsequently, this bibliometric study allowed visualization of the research trends regarding honey’s antioxidant and antimicrobial properties over the last 22 years. The co-occurrence analysis of keywords revealed that research topics such as propolis, flavonoid, anti-inflammatory activity, wound healing, chemical composition, polyphenol, antineoplastic activity, manuka honey, antifungal activity, phenolic compounds, and pollen were highly related to the antimicrobial and antioxidant characteristics of honey.

The initial research focus in this domain was primarily on honey’s anti-inflammatory and antineoplastic activity, wound healing, and antibacterial agents. In the following years, the research agenda was enriched by studies concerning the use of honey and honey products against pathogens such as *E. coli* and *S. aureus*, and the mode of action of propolis, phenolic compounds, oxidative stress, and flavonoids.

Possible limitations of this research might be the single database utilized, the language, the document types, and the selection of specific bibliometric indicators. Despite these limitations, this study provides a significant global output regarding honey research and trends. Overall, there is still a need to understand the mode of action of all antimicrobial and antioxidant agents of honey and honey products, as well as to further investigate all of their potential therapeutic options.

## Figures and Tables

**Figure 1 antioxidants-12-00414-f001:**
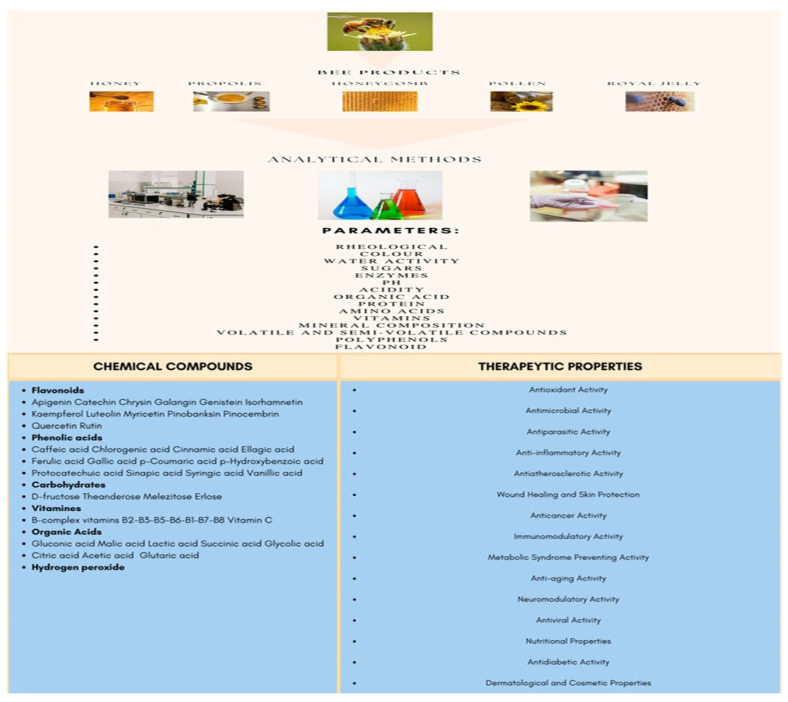
Bee products, chemical composition, and honey therapeutic properties.

**Figure 2 antioxidants-12-00414-f002:**
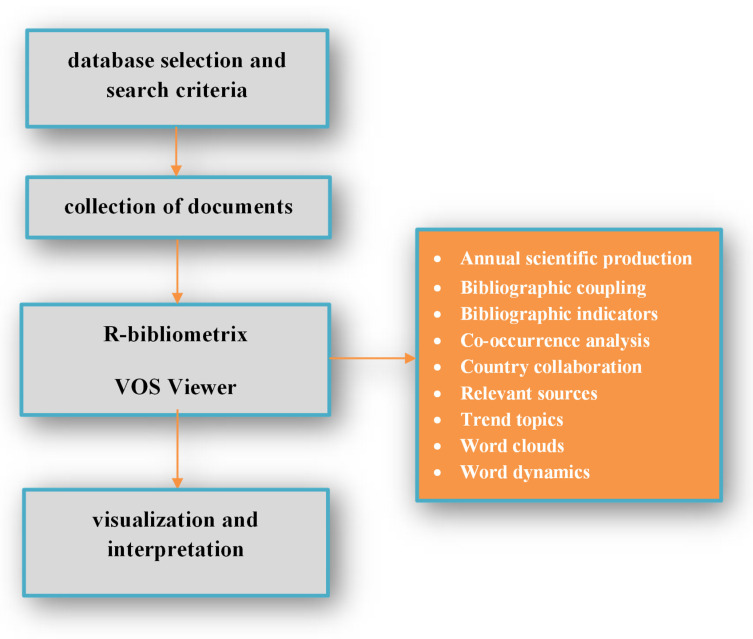
Research flow diagram.

**Figure 3 antioxidants-12-00414-f003:**
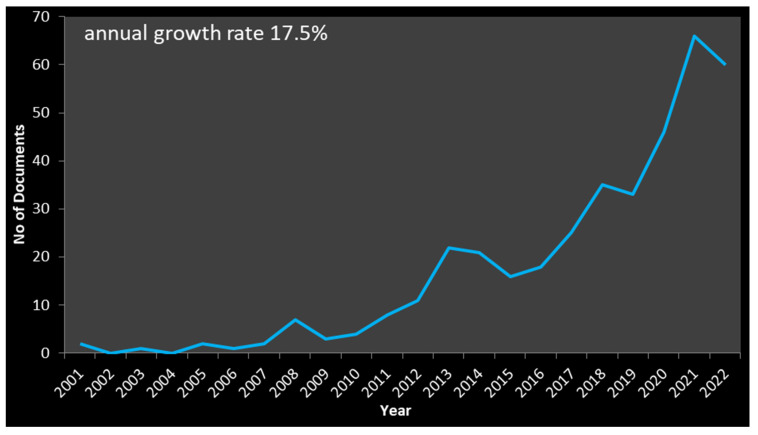
Number of documents per year.

**Figure 4 antioxidants-12-00414-f004:**
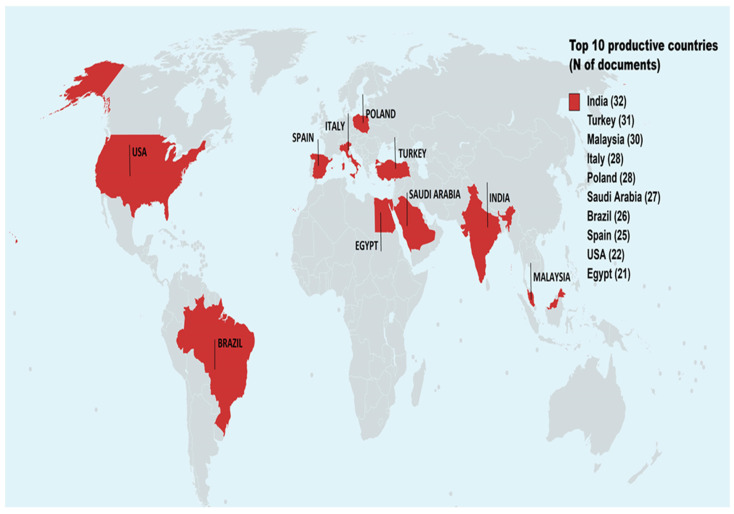
Top ten most productive countries.

**Figure 5 antioxidants-12-00414-f005:**
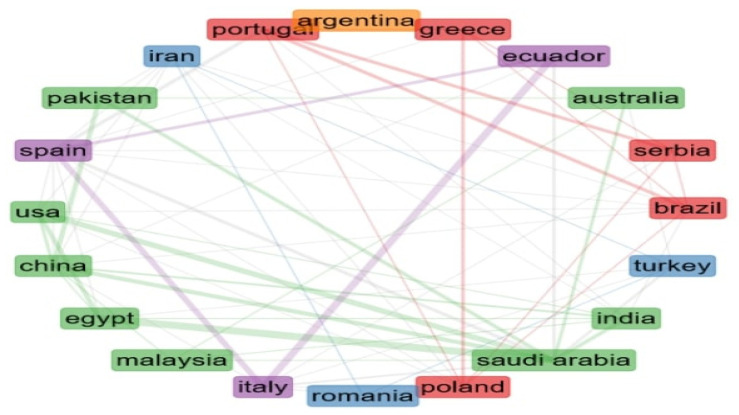
Country collaboration network. (Node size is proportional to the number of publications per country. Edge size is proportional to the number of collaborations between two countries. Nodes are clustered using the Louvain algorithm).

**Figure 6 antioxidants-12-00414-f006:**
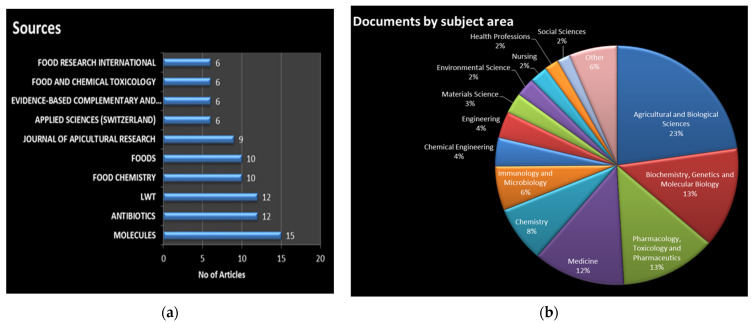
(**a**) Most relevant sources; (**b**) documents by subject area.

**Figure 7 antioxidants-12-00414-f007:**
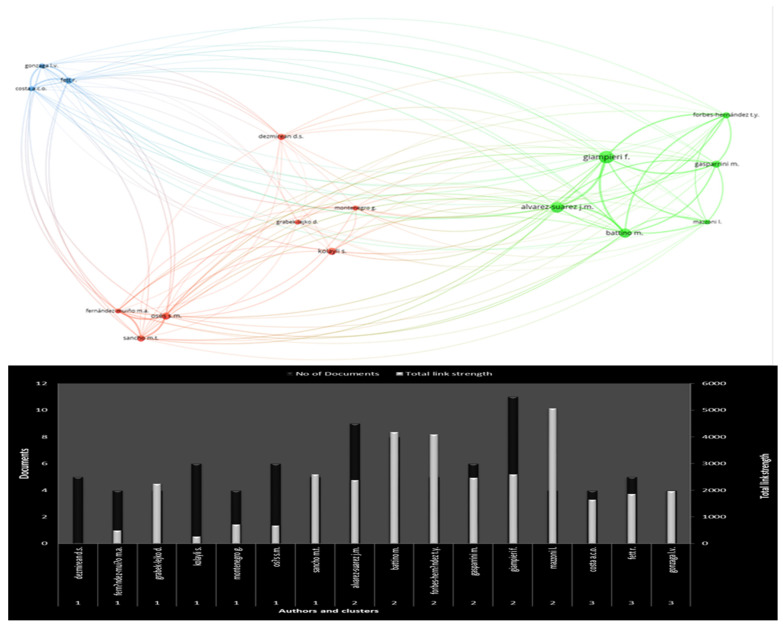
Network, clusters, and number of documents of the authors.

**Figure 8 antioxidants-12-00414-f008:**
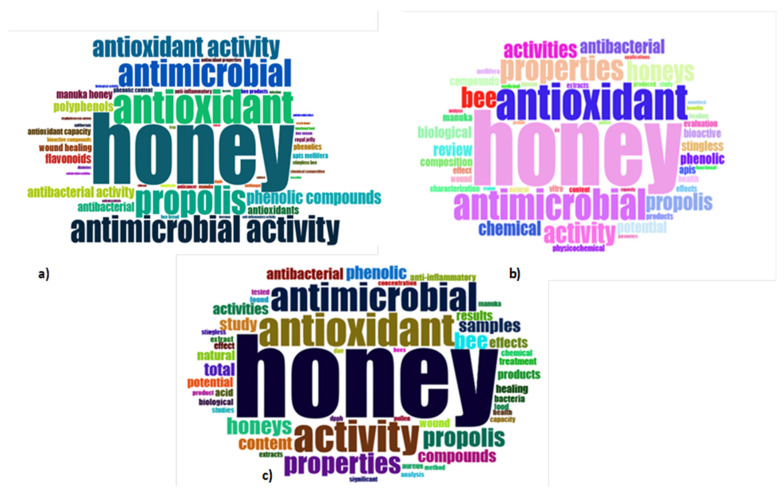
Frequency word clouds of (**a**) titles; (**b**) abstracts; (**c**) keywords.

**Figure 9 antioxidants-12-00414-f009:**
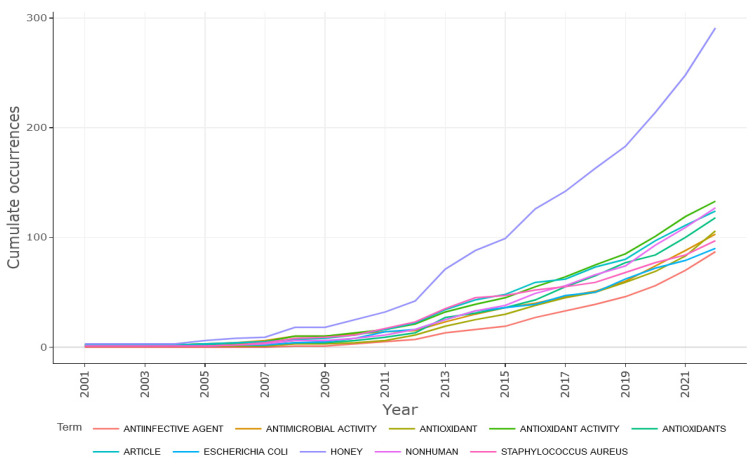
Word dynamics of Keywords plus.

**Figure 10 antioxidants-12-00414-f010:**
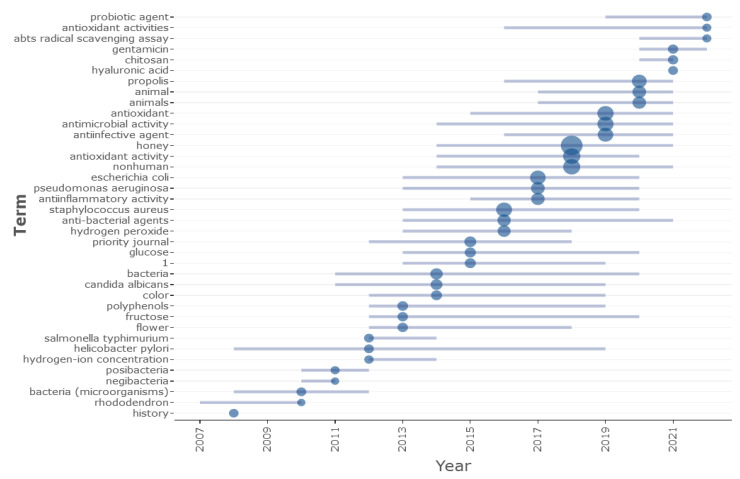
Trend topics of Keywords plus (Interpretation: each topic is represented on the graph by a bubble, while the size of the bubble is proportional to the word occurrences. Minimum word frequency = 5. The grey bar represents the first and third quartiles of the occurrence distribution. 1 = 1-diphenyl 2 picrylhydrazyl).

**Figure 11 antioxidants-12-00414-f011:**
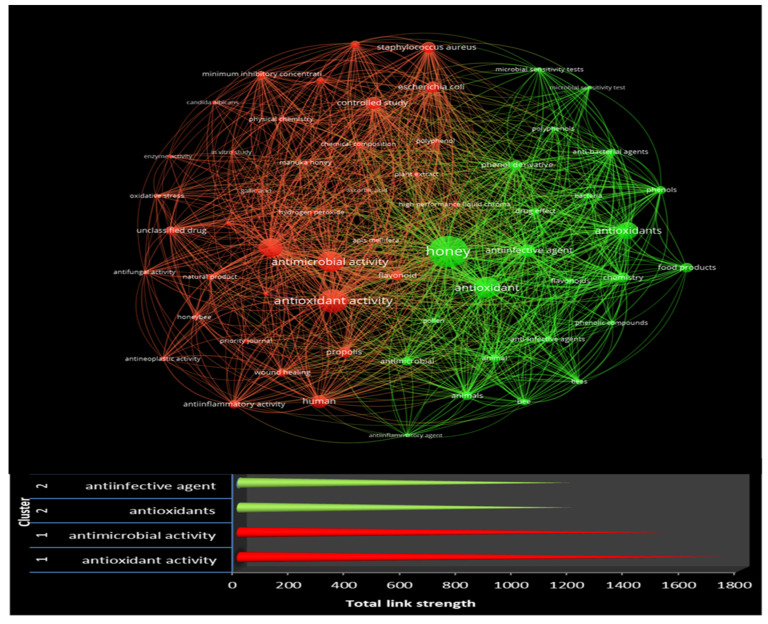
Keyword analysis and total link strength of the keywords in each cluster.

**Figure 12 antioxidants-12-00414-f012:**
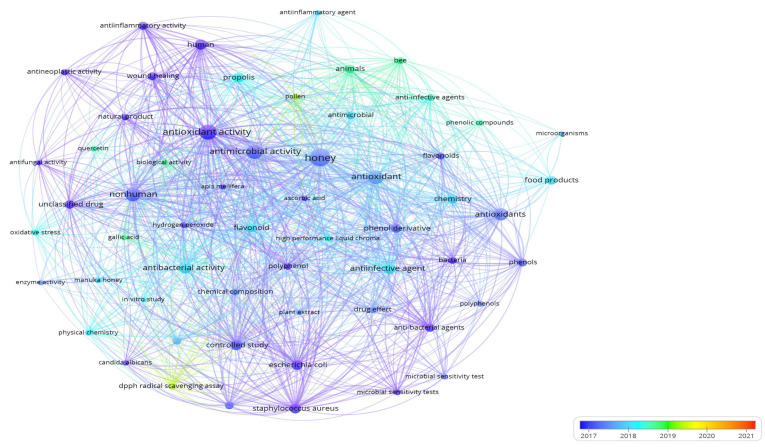
Evolution of the research on the antimicrobial and antioxidant activities of honey, 2017–2021.

**Table 1 antioxidants-12-00414-t001:** Descriptive characteristics of the manuscripts indexed in the Scopus database.

Document by Type N (%)	Top 10 Institutions (N of Documents)	Top 10 Productive Authors (N of Documents)	Authors h Index
Article 273 (71%)	Università Politecnicadelle Marche—Italy (11)	Giampieri, F. (11)	46
Review 81 (21%)	King Abdulaziz University— Saudi Arabia (9)	Alvarez-Suarez, J.M. (9)	36
Book Chapter 19 (5%)	University of Rzeszów—Poland (8)	Battino, M. (8)	70
Conference Paper 8 (2%)	Karadeniz Technical University— Turkey (7)	Gasparrini, M. (6)	34
Conference Review 2 (1%)	University of Belgrade—Serbia (7)	Kolayli, S. (6)	30
	King Saud University— Saudi Arabia (6)	Osés, S.M. (6)	16
	Universidade Federal de Santa Catarina—Brazil (6)	Dezmirean, D.S. (5)	19
	Universidad de Burgos—Spain (6)	Fett, R. (5)	40
	InstitutoPolitecnico de Braganca— Portugal (6)	Forbes-Hernández, T.Y. (5)	38
	Chiang Mai University—Thailand (5)	Sancho, M.T. (5)	23

**Table 2 antioxidants-12-00414-t002:** Top ten most cited articles indexed in the Scopus database.

Author	Title	Year	Source	Cited by
[53]	Honey for nutrition and health: A review	2008	*Journal of the American College of Nutrition* 27(6), pp. 677–689	660
[54]	Antimicrobial strategies centered around reactive oxygen species—bactericidal antibiotics, photodynamic therapy, and beyond	2013	*FEMS Microbiology* *Reviews* 37(6), pp. 955–989	603
[17]	Functional properties of honey, propolis, and royal jelly	2008	*Journal of Food Science* 73(9), pp. R117–R124	552
[55]	Biological activities and chemical composition of three honeys of different types from Anatolia	2007	*Food Chemistry* 100(2), pp. 526–534	370
[56]	Antioxidant and antimicrobial effects of phenolic compounds extracts of Northeast Portugal honey	2008	*Food and Chemical* *Toxicology* 46(12), pp. 3774–3779	359
[44]	Antioxidant and antimicrobial capacity of several monofloral Cuban honeys and their correlation with color, polyphenol content and other chemical compounds	2010	*Food and Chemical* *Toxicology* 48(8–9), pp. 2490–2499	297
[15]	Phenolic compounds in honey and their associated health benefits: A review	2018	*Molecules* 23(9), 2322	257
[57]	Traditional Therapies for Skin Wound Healing	2016	*Advances in Wound Care* 5(5), pp. 208–229	250
[39]	Inhibitory activity of honey against foodborne pathogens as influenced by the presence of hydrogen peroxide and level of antioxidant power	2001	*International Journal of Food Microbiology* 69(3), pp. 217–225	234
[14]	Therapeutic properties of bioactive compounds from different honeybee products	2017	*Frontiers in* *Pharmacology* 8(JUN), 412	207

**Table 3 antioxidants-12-00414-t003:** VOSviewer clusters “antimicrobial-antioxidant-honey”.

Cluster Identification	Keywords
** red **	antioxidant activity, antimicrobial activity, nonhuman, human, controlled study, *Staphylococcus aureus*, *Escherichia coli*, propolis, flavonoid, unclassified drug, antiinflammatory activity, wound healing, minimum inhibitory concentration, *Pseudomonas aeruginosa*, chemical composition, dpph radical scavenging assay, natural product, polyphenol, antineoplastic activity, plant extract, physical chemistry, oxidative stress, hydrogen peroxide, manuka honey, high performance liquid chromatography, biological activity, antifungal activity, apis mellifera, in vitro study, *Candida albicans*, quercetin, ascorbic acid, gallic acid, enzyme activity
** green **	** honey, antioxidant, antiinfective agent, chemistry, phenol derivative, food products, animals, antimicrobial, phenols, anti-bacterial agents, flavonoids, drug effect, bee, anti-infective agents, bacteria, microbial sensitivity tests, phenolic compounds, polyphenols, pollen, microbial sensitivity test, anti-inflammatory agent **

## Data Availability

Data is contained within the article.

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
