# Peer review of "Honey’s Antioxidant and Antimicrobial Properties: A Bibliometric Study"

_antioxidants, 2023, doi:10.3390/antiox12020414_

Round 1
Reviewer 1 Report
This work is a well structured summary or review of the literature focused on the evolution of honey research. It has utility for tracing the origins and manifestations of various properties and hive substances related to honey. In general, Im not sure all the figures are necessary, many of the figures could be simply stated in a sentence. Additionally some of the discovered patterns are socio-cultural and could be eliminated, as its difficult to see their explanatory utility for honey research.
You may also further qualify your use of the Scopus database and search terms, as I dont see many of the major papers I consider influential.
Line 155: Results and Discussion
Line 223: why is this a "discrepancy" or disharmony? isnt it just another pattern revealed by your search terms? Why should they correspond?
Line 453: is this the topic sentence of a new paragraph?
Line 595: you mean "free radical" scavenging?
Line 616: there is no red (2021) evident in figure 11.
Line 647: What does this sentence mean? (Health implications to the hygiene status of humans)
Author Response
Reviewer 1
Comments:
This work is a well structured summary or review of the literature focused on the evolution of honey research. It has utility for tracing the origins and manifestations of various properties and hive substances related to honey.
In general, Im not sure all the figures are necessary, many of the figures could be simply stated in a sentence.
Response:
Primarily, images positively impact the article's readability and enrich the text with accumulated knowledge and information. We comprehend the reviewer's concern but we prefer he indicates which figure he wants to be removed.
Comment:
Additionally some of the discovered patterns are socio-cultural and could be eliminated, as its difficult to see their explanatory utility for honey research.
Response:
We thank the reviewer for that remark and for pinpointing that issue. On the topic of cooperation between countries, we tried to give an additional perspective describing some socio-economic patterns. These lines were deleted:L274-282.
Comment:
You may also further qualify your use of the Scopus database and search terms, as I dont see many of the major papers I consider influential.
Response:
We thank the reviewer for this remark. We have expanded the list of most influential articles by adding five more. Table 5 was reformatted, presenting ten articles. Moreover, we have added some explanations to the text regarding the new articles. Regarding the search strategy, the Scopes database is considered updated and comprehensive. The search performed included general terms to cover as many manuscripts as possible. If some articles are not presented, they will have been excluded mainly due to lack of institutional access or will not be included in the database. In the next round, the reviewer may send us some documents to be discussed and entered into the relevant list.
L332:” Continuing with the other most influential documents, we identified manuscripts that focused on trying to correlate the botanical origin of honey with characteristics such as its colour, total phenolic, flavonoid, ascorbic acid, amino acid, protein and carotenoid contents. Honey's antimicrobial capacity was also tested against Gram-positive and negative bacteria, showing miscellaneous sensitivity results. The correlation between chemical components of honey, such as phenolic content and hydrogen peroxide, influences the general antimicrobial activity of honeys, as documented in the following article in the list presented in Table 2. Dark-coloured honeys achieved an increased inhibitory effect compared to lighter-coloured honeys to various pathogenic microorganisms such as Escherichia coli O157:H7, Salmonella typhimurium, Shigella sonnei, Listeria monocytogenes, Staphylococcus aureus. The following article presented an extensive literature review on the beneficial properties of honey in human health and recorded, in addition to its known antimicrobial and antioxidant properties, beneficial effects on cardiovascular function, the respiratory system, the fight against cancerous tumours and finally, the fight against diabetes. Subsequently, the therapeutic properties of honey and beekeeping products, the characteristic chemical and bioactive substances and their uses even in traditional medicine were the research objectives of the ten most influential articles. “
Line 155: Results and Discussion
Response: Changed. A space was added between the two words.
Line 223: why is this a "discrepancy" or disharmony? isnt it just another pattern revealed by your search terms? Why should they correspond?
Response:
We have rephrased this sentence. We apologize to the reviewer for not explaining this point exactly. We agree that the difference in honey production in some countries and their research in this area varies substantially. One would expect countries with honey in their culture as a nutritional element and an element in their traditional medicine to show increased research activity. However, this pattern is not confirmed in the present research, as stated in the manuscript. The wording and use of the word "disharmony" probably prejudice the reader's position at this particular point, and therefore we proceeded with the following change.
L223:” Thus, diversification is observed in research output and honey production between countries. Honey research can focus on various technological innovations and research directions: antioxidant and antimicrobial properties, medical applications, honey products, entomology and bee biology, product standardization and marketing, and botanical and geographical origin that can justify this detected research pattern.”
Line 453: is this the topic sentence of a new paragraph?
Changed. We rephrased the last sentence of this paragraph and placed it as the first in the following one in line 453: “The map categorized terms into two significant clusters with respective colours. The colour of an item is imposed by the cluster to which the item belongs…”.
Line 595: you mean "free radical" scavenging?
Changed. We thank the reveiwer for than valuable comment.
Line 616: there is no red (2021) evident in figure 11.
Response:
We apologize for this misconception. The year 2021 is included in the colour scale of the map. However, the keywords are not visible on the map precisely because they are presented in more recent articles, which have yet to gain scientific recognition. The most recent manuscripts have been cited less than an article published five years ago. Thus total link strength and average publication year score are deficient. However, VOSviewer in this particular type of analysis and applying this colour scale may show a colour, e.g. red, which is only sometimes visible in the bibliographic map.
Line 647: What does this sentence mean? (Health implications to the hygiene status of humans)
Response:
Changed. We rephrased that sentence and made it more comprehensible. “Honey and honey products are of great importance regarding their nutritional and therapeutic properties and contribute positively and beneficially to human health.”
Reviewer 2 Report
The review "Honey's antioxidant and antimicrobial properties: a bibliometric study" is a well-comprehensive study regarding the publication from 2001 to 2022 indexed in several international databases. The subject is related to an analysis regarding to the honey's nutritional status and its beneficial properties to human health. This study aims to provide a bibliometric analysis of honey's antioxidant and antimicrobial properties.
Some comments:
The authors must considered also the publications which are based on a questionnaire. There are a lot of articles which use the questionnaire in order to evaluate the use of honey and other bee products for pharmaceutical and therapeutic uses.
A graphical analysis or a table which include an analysis regarding the application of honey and its derivatives products must be included.
The references are not in accord with journal requirements: References must be numbered in order of appearance in the text and listed individually at the end of the manuscript.
Author Response
Reviewer 2
Comments:
The review "Honey's antioxidant and antimicrobial properties: a bibliometric study" is a well comprehensive study regarding the publication from 2001 to 2022 indexed in several international databases. The subject is related to an analysis regarding to the honey's nutritional status and its beneficial properties to human health. This study aims to provide a bibliometric analysis of honey's antioxidant and antimicrobial properties.
Some comments:
The authors must considered also the publications which are based on a questionnaire. There are a lot of articles which use the questionnaire in order to evaluate the use of honey and other bee products for pharmaceutical and therapeutic uses.
Response:
We thank the reviewer for that valuable comment. We agree with the reviewer that various articles use a questionnaire to extract and interpret information of high interest in the entire spectrum of research domains related to beekeeping products, honey and their derivatives. In studies about consumers' nutritional habits and behaviour toward these products, clinical studies, and health issues, questionnaires can be applied. In the current bibliometric analysis, four studies included a questionnaire. They referred to consumer acceptance of a honey and propolis product, the organoleptic properties of honey with the addition of various spices, and the application of honey in cases of children's cough. Finally, a questionnaire was designed and given to beekeepers to asses utilized agronomic practices. These manuscripts are depicted in L332.
Comment: A graphical analysis or a table which include an analysis regarding the application of honey and its derivatives products must be included.
Response:
We thank the reviewer for that valuable comment. Indeed, a figure always offers a significant benefit in conveying information. We added a figure in the Introduction section (L54) regarding some essential components of honey and honey products, analytical parameters, chemical compositions of honey and therapeutic properties.
Comment: The references are not in accord with journal requirements: References must be numbered in order of appearance in the text and listed individually at the end of the manuscript.
Response:
We thank the reviewer for this remark. The in-text references and citation list have been reorganized based on journal requirements.
Round 2
Reviewer 2 Report
Dear Authors,
I agree with the publication of the manuscript. The reviewer request were fulfilled.